# Management of Pelvic Pain in Patients with Crohn’s Disease—Current Overview

**DOI:** 10.3390/jcm12020526

**Published:** 2023-01-09

**Authors:** Jakub Włodarczyk, Jacek Burzyński, Bartłomiej Czerwiński, Mateusz Prusisz, Katarzyna Socała, Ewa Poleszak, Jakub Fichna, Kasper Maryńczak, Marcin Włodarczyk, Łukasz Dziki

**Affiliations:** 1Department of General and Oncological, Medical University of Lodz, Pomorska 251, 90-213 Lodz, Poland; 2Department of Biochemistry, Medical University of Lodz, Mazowiecka 6/8, 92-215 Lodz, Poland; 3Department of Animal Physiology and Pharmacology, Institute of Biological Sciences, Maria Curie-Skłodowska University, Akademicka 19, 20-033 Lublin, Poland; 4Chair and Department of Applied and Social Pharmacy, Medical University of Lublin, Chodźki 1, 20-093 Lublin, Poland

**Keywords:** Crohn’s disease, pelvic pain, pain treatment, pelvic manifestations of Crohn’s disease

## Abstract

Crohn’s disease (CD) is a subtype of chronic inflammatory bowel diseases (IBD) with characteristic skip lesions and transmural inflammation that may affect the entire gastrointestinal tract from the mouth to the anus. Persistent pain is one of the main symptoms of CD. This pain has multifactorial pathogenesis, but most often arises from intestinal inflammation itself, as well as from gut distention or partial intestinal obstruction. Some current evidence also suggests sensitization of sensory pathways, as well as modulation of those signals by the central nervous system, which highlights the impact of biopsychosocial factors. To date, most studies have focused only on the pain located in the abdomen, while pelvic pain has rarely been explored, despite it being a common symptom. The aim of this study is to provide an abbreviated summary of the current state of knowledge on the origins and treatment of pelvic pain in CD.

## 1. Introduction

Crohn’s disease (CD) is a chronic, relapsing inflammatory bowel disease (IBD), which can affect any part of the gastrointestinal tract; however, the ileum, colon or both are the most commonly involved areas [1]. In addition, CD is often associated with a number of different extraintestinal manifestations that also significantly affect the course of disease as well as greatly impact patients’ quality of life (QoL). The prevalence of CD ranges from 3 to 20 cases per 100,000 [2] and is more common in the industrialized world, including Western Europe and North America [3], though the number of CD patients is on the rise in Asian developing countries. CD is mostly associated with abdominal pain and diarrhea, which significantly impair everyday activities of affected patients. More systematic symptoms such as weight loss, low-grade fever and fatigue are also commonly reported. Sometimes intestinal strictures may develop in the course of the disease, leading to subileus or ileus with signs of abdominal cramps. 

Although pelvic pain is not widely regarded as the principal symptom of IBD, in some patients it may not only be present, but even constitute their primary complaint. Pelvic pain generally occurs under the navel or as low back pain, and may exist with or independently from abdominal pain. It can also occur as rectal, vaginal, or perineal pain, typically in the course of perianal Crohn’s disease (PCD). The existing literature has primarily focused on abdominal pain in CD patients, while pelvic pain remained largely unexplored, even though both may significantly impair QoL, leading to serious mental disorders, including depression and anxiety [3].

This paper will discuss pathophysiology and a multidisciplinary approach to pain management in CD patients with special emphasis on disease-related pelvic pain. The main trends and recent reports on development of future treatments are discussed herein as well.

## 2. Brief Main Headings

### Pathogenesis of Pain in Crohn’s Disease

Pain in CD arises from a complex mechanism and can be regarded as a marker of progression of the disease. Furthermore, there is a noticeable increase in the mortality of those CD patients who require long-term use of analgesics for management of chronic pain [4]. Pain intensity varies among the patients, depending on phenotype and advancement of disease, and is influenced by individual and psychosocial factors, such as stress, hypersensitization or genetic burden. This complexity underlines the need for a multidisciplinary approach in order to achieve proper pain management in CD patients.

The Montreal classification describes three phenotypes of the CD: penetrating, stricturing and nonpenetrating, and nonstricturing. The latter is mainly associated with inflammation affecting all layers of GI wall, which is exhibited by characteristic skip lesions. Then, with the advancement of disease, the remaining two phenotypes may develop. The stricturing phenotype is characterized by the deposition of excessive connective tissue in the GI wall, which may lead to disturbances in ileal passage and eventually obstruction. In contrast, penetrating CD is exhibited mainly by formation of fistulae, ulcers or abscesses [5]. 

There is little data concerning the relation between particular phenotypes of CD and the level of pain sensations. It can be assumed that in the context of pelvic pain in CD, special attention should be given to penetrating phenotypes. Depending on localization, some fistulae or abscesses may result in greatly aggravated sensations of acute pain. In the course of stricturing CD, especially in case of bowel obstruction, patients often present with painful abdominal cramps, although pelvic pain may occur as well [6,7]. 

Pain in CD arises mainly through visceral mechanisms. Noxious or any pain-producing stimuli are detected by nociceptors—specialized primary afferent neurons located in the wall of the gastrointestinal tract (GI) [8]. The polymodality of these receptors allows them to react to a wide array of stimuli, such as thermal, chemical or mechanical. Afferent endings of the GI tract are characterized by the ability to change their excitability in response to various pro- and anti-analgesic agents [9]. A prolonged inflammatory state in CD may lead to hypersensitization of tissue and result in chronic abdominal or pelvic pain [10]. The exact mechanism of this phenomenon is still unknown. It is believed that inflammation-related hypersensitivity arises as a result of changes in mucosal and nociceptive properties, along with modulation of pain perception in the central nervous system or inhibition of sacral dorsal horn neurons [10]. An alternative hypothesis highlights the observation that the immune system is a critical determinant of visceral nociception and that T lymphocytes have significant opioid-mediated antinociceptive influence in the gut [11]. 

Sensory neurons may be impacted by several mediators, such as serotonin, histamine or prostaglandins, which are released by enteroendocrine cells. Recent studies revealed that CD patients suffering from hypersensitivity exhibit an increase in the number of enterochromaffin cells (ECCs) and mast cells in both inflamed and healthy tissue. Moreover, ECCs and mast cells are involved in the regulation of gut motility and may be responsible for the development of functional gut disorders [12,13].

Changes in the expression profile of nociceptors themselves were observed as well, especially in transient receptor potential cation channels (TRPV1)-induced hyperalgesia. TRPV1 channels are present in the membrane of nociceptors and are involved in pain sensations. A significant increase in the amount of TRPV1 channels and their excitability was observed in CD patients and was associated with elevated proalgesic factors, such as NGF (nerve growth factor) and GDNF (glial cell line-derived neurotrophic factor). Effectiveness of antibodies against NGF and other neurotrophic factors in treatment of chronic pain is currently being examined. Up-regulation of N-methyl-D-aspartate on second-order neurons was also associated with hyperalgesia and along with other factors may be present even after resolution of inflammation, contributing to the maintenance of symptoms [14].

fMRI (functional magnetic resonance imaging) and PET studies on pain suggest a significance of the central nervous system (CNS) in hyperalgesia development, mainly by downregulation of inhibitory descending pathways (exhibiting analgesic function) and facilitating excitatory synaptic responses in pain fibers [12].

Sociopsychological factors along with mental disorders, including high stress and anxiety levels or depression, may contribute to hypersensitivity. This idea is supported by a study by E. Garica-Vega and C. Fernandes-Rodriguez, which revealed that CD patients participating in stress management programs, while receiving conventional medical treatment simultaneously, had reduced level of pain sensation and frequency of diarrhea as compared to the control group [15]. Recent studies on the brain–gut axis in IBD revealed the presence of stress-induced perturbations leading to the development of inflammation and disease progression. It was observed that stress inhibits anti-inflammatory properties of the vagal nerve, whereas the proinflammatory influence of the sympathetic nervous system is amplified. Moreover, research data suggest gut microbiota involvement in development of IBD, and brain–gut axis may have mediatory function in that respect as well. Stress-induced increase in permeability of GI tract allows bacteria to cross epithelial barrier and activate local immune responses, and that in turn triggers inflammation of the mucus [16].

## 3. Introduction to Pelvic Pain

Pelvic pain, defined as any pain sensations located beneath iliac spines, can constitute a difficult problem to properly diagnose and treat, because perceived pain sensation originating in the pelvic wall and viscera is rarely associated with this area. The major complaint of patients with CD located in the terminal part of GI or with extraintestinal manifestations affecting the pelvic viscera and wall is most often not pelvic pain, but abdominal pain. Pelvic pain alone in CD is rarely mentioned in the literature and typically is associated with conditions such as PCD, metastatic CD located in the vulva or fistulae between GI and pelvic wall. This scarcity may be partially explained by erroneous merge of terms “pelvic pain” and more common “abdominal pain”, since both often share similar traits and may occur simultaneously. 

Notably, pelvic and/or abdominal pain may persist even after successful treatment and resolution of all detectable causes (i.e., inflammation, abscess or constriction). It is thought that several processes may be involved in pathogenesis of this chronic pain, for example visceral hypersensitivity, disrupted regulation of pain signaling and neurotransmitter release, modulation of signals in the central nervous system, psychological effect [13]. Summary of the pathogenesis of pelvic pain in CD can be found in Figure 1.

### 3.1. Pelvic Anatomy

Pelvic anatomy plays a crucial role in the complex pathogenesis of the pelvic pain in CD. While rectum may be affected in both females and males during CD progression, the female pelvis contains a few additional structures in which extraintestinal manifestations of CD may occur. Uterus and perianal area involvement are commonly reported, and some patients may also exhibit vulvar and cystic manifestations.

Innervation of pelvic components comes from an intricate neuronal system, formed mainly by branches of inferior hypogastric plexus, along with pelvic and sacral splanchnic nerves. Sympathetic fibers originate from the thoracolumbar spinal cord, whereas parasympathetic nerves come from the sacral part. Although both are involved in nociception, there is a noticeable dominance of sympathetic components. The complexity of pelvic innervation along with atypical entrance of sacral afferents to CNS (through ventral horns rather than dorsal [17]) result in an inability to pinpoint the sensations to a particular area. As a result, patients with perianal manifestation or urogenital manifestation often report poorly localized pain, diffused over the lower parts of the abdomen. 

### 3.2. Perianal Crohn’s Disease

PCD, although a common occurrence, is often considered a complication rather than a separate phenotype of the disease. It is estimated that PCD symptoms are present in 18–43% of CD cases and may lead to severe pain and impairment of QoL [18]. PCD encompasses fistulizing (abscesses, fistulae) and non-fistulizing (fissures, stenosis, ulcers and skin tags) variant [19]. Whereas stenosis and skin tags, if not inflamed, are considered painless, ulcers, fissures, abscesses and fistulae are strongly associated with pain and discomfort [19,20].

Fistulizing PCD manifestations, including anal fistulae and abscesses, are often associated with pain. Anal fistulae or fistulae-in-ano, abnormal connections between rectum with other organs (especially vagina, vaginorectal fistula) or skin around the anus, are one of most disabling manifestation and occurred with an incidence up to 50% of PCD cases [21]. Patients usually report purulent discharge, intermittent perianal edema, different levels of pain and stool incontinence. The mechanism of fistulae formation is still unknown, but it seems that inflammation of rectal sweat glands and perforating ulcers may play a key role. Abscesses may form as a result of cavitating lesion or cryptoglandular infections and are associated with symptoms such as persistent perianal pain, fever, night sweats and chills [18,22].

Recently, the impact of the microbiome on CD-associated fistula development and recurrency has been highlighted by research investigators [23,24]. The main rationale supporting the significant role of bacterial contribution in etiopathology of fistulae is the efficacious role of antibiotics in perianal fistulae treatment. It was observed that mainly Gram-positive microorganisms (i.e., Corynebacterium) were associated with perianal fistulae tract samples. A possible mechanism involves bacterial cell wall muramyl dipeptide, which stimulates the expression of molecules linked with EMT (TNF-α, TGF-β, SANIL1, and IL-13) [25]. Moreover, some bacteria were reported to activate the onset of ENT directly (Citrobacter rodentium, E.Coli) [26,27]. 

Non-fistulizing variant of PCD pain is commonly reported. Half of patients suffering from cavitating ulcers (erosion of rectal wall, sphincter muscle or perianal tissues) report remittent and severe pain. Classically, anal fissures were considered to be painless, but new reports show that 44–70% of PCD patients suffer from anal discomfort including pain [19]. Moreover, some fissures may transform into painful ulcers or fistulae. Uninflamed skin tags and rectal stenosis are rarely associated with pain [19,20].

### 3.3. Metastatic Crohn’s Disease and Gynecological Manifestations of CD

Metastatic Crohn’s Disease (MCD) is one of the most challenging diagnostic problems due to its rarity and lack of awareness of medical practitioners of the existing association between Crohn’s disease and MCD [28]. Metastatic manifestations are described as a granulomatous cutaneous lesion separated from the affected gut by healthy tissue [29]. Moisture of area increases risk of MCD formation [30]. In regard to pelvic pain, the vulvar MCD is the most prominent. Patients affected with this variant usually report chronic vulvar pain and edema (usually inflammatory and asymmetrical; present in 67% of cases), knife-like ulcers, commonly extending to groins (present in 40% of cases) and chronic suppuration [29]. The direct process underlying vulvar manifestations in CD patients is still unclear. 

Gynecological manifestations other than MCD are generally quite common in female CD patients. In the majority of cases, menstrual abnormalities, including dysmenorrhea and heavy bleeding, are present during CD flares. Dyspareunia and impairment of sexual life may also occur [31,32]. A multidisciplinary approach as well as increased awareness among gynecology and gastroenterology providers of coexistence of CD and MCD in female patients are needed.

### 3.4. Functional Gastrointestinal Disorders in CD

Functional gastrointestinal disorders (FGID) are a broad group of syndromes, including gut motility disturbances and sensitivity abnormalities, which can occur independently, but for the purpose of this review, only the comorbidity with CD will be discussed.

Patients suffering from CD often develop FGID, mainly gut malfunction leading to defecation disorders and IBS. As a result, there is a reduction in colon contractility and the number of spontaneous contractions. The role of these disturbances in the development of diarrhea is still unclear, but it is believed that motility abnormalities along with secretory and resorption impairment can be responsible. Diarrhea is often accompanied by abdominal and/or pelvic pain. The FGID symptoms may occur in areas distant to the primary inflammation site, i.e., CD patients with involvement of ileum, who suffer from gastroparesis and decreased oroceacal transit. Furthermore, FGID may contribute to development of malnutrition and anorexia, which itself impacts QoL and complicates treatment [13]. 

Interestingly, it was reported that IBD patients are at higher risk of developing irritable bowel syndrome (IBS)-like syndromes, even despite clinically and endoscopy confirmed remission of disease. IBS is a chronic disorder, whose symptoms, i.e., abdominal pain and defecation disorders, fluctuate in a period of time. It has been suggested that IBS arises from motility disturbances and visceral hypersensitivity, although the exact pathology underlying this dysfunction is still unknown [13].

The most recent meta-analysis revealed that 36.6% of patients in CD remission suffer from IBS-like symptoms, including pain [22]. The cause of this phenomenon is unknown, but it was suggested that subclinical inflammation, dysbiosis, disturbed gut barrier or hypersensitivity of visceral nociceptors may contribute to maintenance of IBS-like symptoms after inflammation resolution [33,34].

## 4. Management of Pelvic Pain in Crohn’s Disease

Contemporarily, there is no definitive, universal treatment for pain stemming from the CD’s activity. Available pharmacotherapy includes classical analgesics agents such nonsteroidal anti-inflammatory drugs (NSAIDs) and opioids, along with new therapeutic strategies, e.g., biological treatment. Surgical procedures are indicated mainly, when more serious manifestations develop. Recent studies emphasize the importance of psychotherapy and a holistic approach to the management of pain in CD [15]. 

### 4.1. Pharmacology

In pain associated with exacerbation of CD, the conventional approach will often be an escalation of therapy aimed at suppressing the inflammatory response. Rapid symptom relief is commonly achieved with steroids or anti-TNF, and thiopurines or methotrexate are instituted for long-term maintenance [35]. When establishing a treatment plan, an activity of the disease and site of lesions should be taken into consideration in order to provide adequate treatment. In some patients, the pain may persist despite therapy and consequently the use of additional analgesics, such as opioids or NSAIDs, may be required. There is no pharmacotherapy specifically targeted at pelvic pain in CD. Standard treatment usually involves a combination of analgesics and anti-inflammatory agents [36].

Pelvic and abdominal pain and pain associated with extraintestinal manifestations (e.g., spondyloarthritis) are often treated with NSAIDs. NSAIDs are easily available without prescription, so many patients are using them for pain management. This class of drugs inhibit the production of prostaglandins by cyclooxygenase (COX) enzymes. While COX-2 is a key mediator of inflammatory pathways, COX-1 is a constitutive enzyme with homeostatic functions, for instance maintaining mucosal integrity of the intestine. Traditional non-selective NSAIDs act by inhibiting both COX-1 and COX-2, thereby often resulting in GI side effects (e.g., inflammatory intestinal lesions) in case of prolonged use. Some studies have demonstrated that NSAIDs may exacerbate IBDs, causing a flare and increasing activity of the disease [36]. Nevertheless, it should also be emphasized that IBDs are heterogeneous in nature, and therefore, only some subgroups of patients may be especially at risk of these adverse effects [37]. Due to limited research data on the effects and safety of long-term use of NSAIDs in IBDs, their use should for now be restricted or avoided altogether, with the exception of patients with debilitating arthritis that cannot be controlled by other means, in which case use of selective COX-2 inhibitors may be considered [38].

Opioids, which act on opioid receptors to produce morphine-like effects, are frequently used to control severe acute pain that cannot be diminished with less powerful drugs. Opioid receptors are primarily associated with the regulation of pain, along with addictive behaviors. Due to these properties, opioids are burdened with a great threat of abuse. One study demonstrated that narcotics are used in 70% of hospitalized patients with IBD [36]. Another study concluded that 30% of all patients with IBD have been prescribed opioids during the years 2010–2013, and that number has been on the rise [39]. Furthermore, in another study, it was observed that within 10 years of diagnosis, 5% of individuals with IBD had become heavy opioid users. The use of opioids at low or moderate intensity before diagnosis is strongly associated with heavy use of opioids later in the course of disease [40,41]. Psychological evaluation and close monitoring by physicians are strongly indicated in patients deemed to be at high risk for opiate abuse, especially in those with symptoms of clinical depression.

Unfortunately, aside from potential opioid use disorder, prolonged use of opiates is often associated with several other side effects such as nausea, respiratory depression, sedation, and euphoria/dysphoria; most of which usually subside with time [42]. Constipation, which could at first glance be regarded by a patient as a relief due to diarrhea being a major symptom of CD, in fact may be dangerous in patients who are at risk of developing toxic megacolon [4]. Opioids may also cause narcotic bowel syndrome (NBS). This syndrome is identified by a chronic, progressive abdominal pain of an intensity that worsens with escalating doses of opioids [4]. Thus, recognizing NBS is important in order not to cause a vicious circle by repeatedly increasing the dose in an attempt to control a patient’s pain.

Prolonged opiate use may also induce bowel hyperalgesia through one of three proposed mechanisms: existence of bimodal opioid regulation system and preferential activation of excitatory pathways over time, which in turn would cause opiate tolerance and pain intensification; the existence of counter-regulatory systems, which would secrete anti-opioid neuromodulators; and glial cell activation, which produces morphine tolerance and increases opiate prompted pain [43].

Use of opiates was also linked to increased susceptibility to infection due to decreased gut motility, which allows bacteria translocation through inflamed mucosa [4]; or due to possibly having direct immunosuppressive effects [44]. Moreover, opiates may be masking early signs and symptoms of infection.

Antidepressants may be used as an “adjuvant analgesic” to reduce chronic opioid intake. Unfortunately, much evidence for their use in IBD originates from studies of IBS [4]. Even at low doses, tricyclic antidepressants seem to decrease the severity of pain in patients with IBD. This effect could be attributed to modulation of abdominal pain perception in the CNS [45]. Furthermore, animal studies have demonstrated anti-inflammatory potential of tricyclic antidepressants, although further trials are required [46]. Selective serotonin reuptake inhibitors (SSRIs) seemingly do not alleviate pain directly, but may improve it secondarily through treating anxiety and depression that may be present in 20% of patients with IBD [14,46].

Recently, anticonvulsants such as gabapentin and pregabalin have been used to treat visceral pain and some studies have shown their beneficial effect on visceral hypersensitivity [47,48]. Gabapentin may also have anti-inflammatory properties through its interaction with the PPAR-gamma receptor, which is the main regulator of NFκB, an activator gene that plays a key role in intestinal inflammation [49]. Further clinical studies are required in order to establish these drugs’ role in management of pain in IBDs.

Overview of pharmacological treatment can be found in Table 1.

### 4.2. Surgical Approach

Unfortunately, despite the development of aforementioned pharmacological therapies, the number of patients will still undergo a surgery within 10 years of CD diagnosis, and many will require subsequent interventions [46]. Surgical approach in pelvic pain in patients with CD should be considered in case of intestinal obstruction, formation of an abscess or fistula, perforation, hemorrhage, perianal skin lesions and anal canal lesions. Many of those can manifest as severe pelvic pain, and surgical intervention may greatly improve patients’ long-term QoL. The most common indication for surgery in patients with disease localized primarily to the colon is failure of medical therapy. Still, surgical approach strongly depends on the exact etiology of pain and therefore it is not possible to suggest a treatment, which would be suitable for everyone, or to guarantee that it would actually resolve this individual patient’s pain. 

For example, if the disease is confined to the right colon, a right hemicolectomy is indicated; if the disease involves the transverse colon or extends to the descending colon, an extended right hemicolectomy or an abdominal colectomy will be needed; in the presence of pancolitis, a proctocolectomy with end ileostomy is the procedure of choice. In general, patients who undergo a total colectomy or total proctocolectomy with ileostomy have lower recurrence rates than those undergoing a segmental colectomy [50], and thus may help avoid recurrence of pelvic pain as well. The advantage of segmental colectomy, however, is that it does not necessitate a creation of stoma and retains bowel function. While the impact of stoma on everyday activities of a patient always needs to be taken into consideration, patients of advanced age may be more tolerant of a permanent stoma than younger patients. Both laparoscopic and open surgery approaches may be used for this procedure. Among notable indications for complete proctocolectomy and end ileostomy are refractory proctitis, pancolitis, perianal fistula or abscess formation, and anal incontinence.

Although the procedure may provide pain relief and therefore greatly improve the patient’s QoL, several issues have to be taken into consideration. Healing of perineal wounds is sometimes problematic, and in up to 10% of patients an infection of the surgical site may develop. A rare complication is an incidental injury to pelvic nerve, which may cause incontinence or/and sexual dysfunction [51].

The most common indications for surgery in PCD are septic in nature (abscesses and fistulae). While anal fissures are typically treated by pharmacological means, and fissurotomy should be considered mainly in refractory cases, both perianal fistula and abscess often require incision, drainage and placement of a seton [52]. Rectovaginal fistulae may occur as a complication of anorectal Crohn’s disease in about 10% of patients and are typically associated with a deep rectal ulceration. 

There is no medical therapy which would reverse the fibrostenotic changes that may cause strictures. In those cases, the goal of the surgery is the resection of the affected part of the bowel. 

### 4.3. Psychotherapy

To date, management of IBD has mainly focused on surgery or medications that decrease inflammation, without addressing the psychological burden of the disease and its influence on the perception of pain; nonetheless, psychological treatment should not be neglected, as it has been shown that it may improve patient outcomes in GI disorders that can be exacerbated by stress or emotional symptoms. Psychosocial interventions such as cognitive behavioral therapy (CBT), hypnosis, and mindfulness techniques have demonstrated the most promise for pain management in IBD. Therefore, psychotherapy may be encouraged as a complementary therapy in patients as a means to learn to cope with chronic pain and psychological burden of fluctuating CD symptoms. Psychotherapeutic intervention is especially recommended for patients who are chronically using opioids [53].

Avoidance-related emotions, thoughts, and behaviors for patients with chronic pain are associated with negative emotions, catastrophic thinking, and inhibition of activity, which in turn leads to increased perception of pain. The goal of behavioral treatments is to teach the patient to change maladaptive thought (e.g., worrying) and behavior to improve mood and bodily responses so that the focus on painful stimuli is reduced. Another approach is to put emphasis on focusing awareness on the present moment, acknowledging and accepting one’s thoughts, behaviors, and bodily sensations without judgment. Both targeting pain-specific thoughts and behaviors such as pain catastrophizing, as well as improving pain acceptance and resilience, may improve a patient’s psychological well-being, mood and perception of pain [54].

Although psychotherapy may result in some improvements in anxiety and depression, no direct effect on the disease activity itself has been noted [4].

### 4.4. Cannabis—Therapeutic Potential

From many years cannabis was considered as only a recreational drug, which affects sensory perception and results in euphoria [55]. Lastly, cannabis has gained medical significance and is being implemented as a treatment for multiple diseases, such as glaucoma, autoimmune diseases, cancer and also Crohn’s disease [55,56]. Cannabis consists of various cannabinoids, where the two mains are cannabidiol and delta-9-tetrahydrocannabinol (THC). They act through the cannabinoid receptors, CB1 and CB2, which can be found robustly in the immune cells, nervous system and gastrointestinal tract. Cannabis is considered to affect inflammation, secretory response and motility in gastrointestinal system [57]. 

Storr et al. questioned 313 patients with IBD, who self-administrated cannabis. Overall, 83.9% of patients reported decreased abdominal pain. However, cannabis use was connected with an increased risk of surgery in patients with Crohn’s disease [58]. In Germany, 417 patients with IBD answered to questionnaire about use of cannabis. Patients who used cannabis reported improved sleep quality, reduced abdominal pain, and relief of worry. Nevertheless, in comparison with non-users, they also reported higher levels of anxiety and decreased quality of life. More than half of the patients received the cannabis from the black market [59]. In New Zealand, 334 surveyed IBD patients reported improvement in loss of appetite, and alleviation of abdominal pain, nausea and vomiting [60]. Despite the various benefits of use of cannabis, especially among IBD patients, studies of a higher evidence level should be conducted.

## 5. Future Perspectives

Increasing understanding of the nature and mechanisms of pain, along with raising knowledge of CD pathology and its manifestations, result in a more holistic approach to management of pain in CD patients and development of new therapeutic strategies. Recent research investigating pharmacological treatment of pain and inflammation in IBD revealed the high potential of new agents, such as agonist of nociceptive receptors, TRPV4 channels antagonists, Janus Kinase (JAK) inhibitors or bioelectrical stimulation of the vagal nerve [61,62].

Nociception receptors (NOPs) are expressed on muscle cell membranes and neurons in the GI tract. They are stimulated by nociceptin, an endogenous agent, which is involved in the regulation of pain signaling and modulation of neurotransmitter release. Experimental studies on GI inflammation models report the presence of alteration of nociception system, which suggests its involvement in pathophysiology of IBDs. Applying NOPs agonists on mice led to an antinociceptive effect; however, evidence of similar results in the human body is lacking. Recent research conducted on non-human primates suggests that NOPs and Mu opioid peptide (MOP) receptor agonists could have analgesic properties without opioid side effects and addictive potential [61].

New therapeutic strategies are also aimed at TRP channels, which are associated with hypersensitization and either inflammatory or anti-inflammatory effects. Activation or inhibition of specific TRP channels may contribute to the occurrence of inflammation in IBD and consequently to perception of pain. While TRPV1 exhibits anti-inflammatory activity by decreasing the production of pro-inflammatory cytokines, stimulation of TRPV4 activates NF- kB, the most important transcription factor during inflammation. Therefore, TRPV4 antagonists may be a promising IBD therapy. It was recently evidenced by Fichna et al. that TRPV4 antagonism alleviates colitis and abdominal pain, paving a pathway to potential novel anti-inflammatory drugs in the future [63]. 

Recent opioid peptides strategies focus on finding opioid agents characterized with high selectivity, efficacy, low side effects and especially resistance to proteolytic degradation in GI tract and serum, which so far hindered their clinical implementation. It is achieved by modification introduced into natural opioids, such as insertion of unnatural amino acids and covalent or non-covalent constraints, cyclization, and design of peptidomimetic ligands, glycopeptides, and N-terminal amidinationed analogs.

There is also a possibility for development of plant-derived opioid-targeting compounds. For example, PR-38, an analogue of salvinorin A from *Salvia divinorum*, has shown anti-inflammatory properties via cannabinoid type 1 receptors and reduced abdominal pain in animal models through μ-opioid receptors [64].

Among potential procedural treatments, nerve blocks have been proposed as a means of management of persistent pain in patients with no signs of bowel inflammation. Pudendal nerve blockade has been successfully used for treatment of chronic pelvic pain and neuropathic pain of gynecological origin [65] and celiac plexus nerve blocks have been used for many years as a treatment for pain associated with pancreatic cancer and chronic pancreatitis [66]. There is a possibility that this method could be utilized in IBDs, though so far there is no research data to support it. Furthermore, it is noteworthy that pain in IBDs is complex and varied in origin, and it is unknown if blocking one specific nervous pathway could be efficient.

Almost 40 years ago, transcutaneous electrical nerve stimulation (TENS) was used for treatment of functional abdominal pain in an open trial of 29 adults [67]. In 2009, a study of patients with functional dyspepsia found improvement in abdominal pain after TENS compared with placebo [68]. The exact mechanism is not yet fully understood. Decrease in spinal N-methyl-D-aspartate receptor phosphorylation was noticed in animal models with visceral hypersensitivity [69]. There is no research data on efficacy of TENS for inflammatory or irritative abdominal pain. 

Differences in gut microbiota composition between healthy individuals and patients with IBD, and the significant role the microbiota play in its pathogenesis, turned the attention of researchers to potential use of fecal microbiota transplant (FMT) in the treatment of IBD. Patients with IBD have a higher ratio of pathogenic bacteria to commensal flora, and bacterial invasion of mucosa is a common occurrence [70]. A systematic review of 18 studies (122 IBD patients) using FMT reported a 61% remission rate in CD patients’ subgroups [71]. Several adverse events are associated with FMT, such as mild fever and mild GI symptoms, which resolve spontaneously. FMT appears to be relatively safe, but variably efficacious treatment for IBD and more randomized controlled trials are needed.

There is also growing support for the use of psychological means to reduce the need for prolonged opioid use and improve general well-being of the individual patient. Psychosocial care training programs for social workers, remote monitoring, and use of telemedicine to provide patients with regular contact and reassurance are among some of the most feasible and potentially beneficial options that are yet to be implemented. A stepped-care model, composed of social workers, psychologists and psychiatrists, may also be used to better address the diverse psychological needs and problems of patients with CD [54]. 

## 6. Conclusions

Understanding of pathophysiology of CD and its applications to various therapeutic approaches have greatly improved over the past decades. However, despite the wide range of available treatments, management of abdominal and pelvic pain still remains a significant challenge for both CD patients and medical practitioners. Mechanisms of pain in CD are not yet fully understood. According to current research, it originates from interaction between sensitization caused by inflammation and modulation of input by CNS. The primary approach to treatment of pain, both abdominal and pelvic, is pharmacological or surgical therapy aimed at the reduction of inflammation and other perceptible manifestations of CD. Not without significance is the poorly understood chronic form of pain, which does not subside despite clinically confirmed resolvement of the inflammation. In order to better understand its pathophysiology, as well as improve the effects of treatment and consequently quality of life of CD patients, further research is needed, as well as holistic and multidisciplinary approach in clinical setting.

## 7. Methodology

We performed research of current literature published on PubMed, Scopus and Google Scholar. No inclusion criteria with regard to the period of publication were used. The research strategy included the following terms: “Crohn’s disease”, “pelvic pain”, “pelvic manifestations”, “perianal Crohn’s disease”, “vulvar manifestations”, “metastatic Crohn’s disease”, and “chronic pain”. The selection of eligible papers is illustrated in Figure 2.

## Figures and Tables

**Figure 1 jcm-12-00526-f001:**
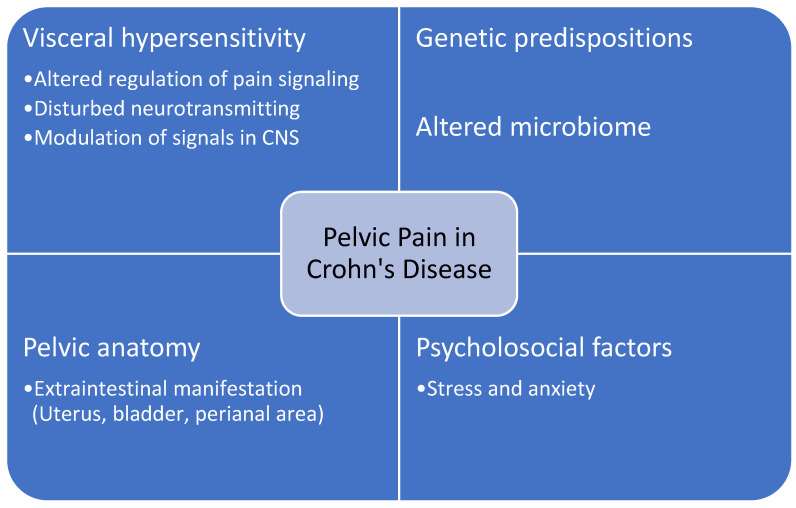
Summary of the pathogenesis of pelvic pain in CD.

**Figure 2 jcm-12-00526-f002:**
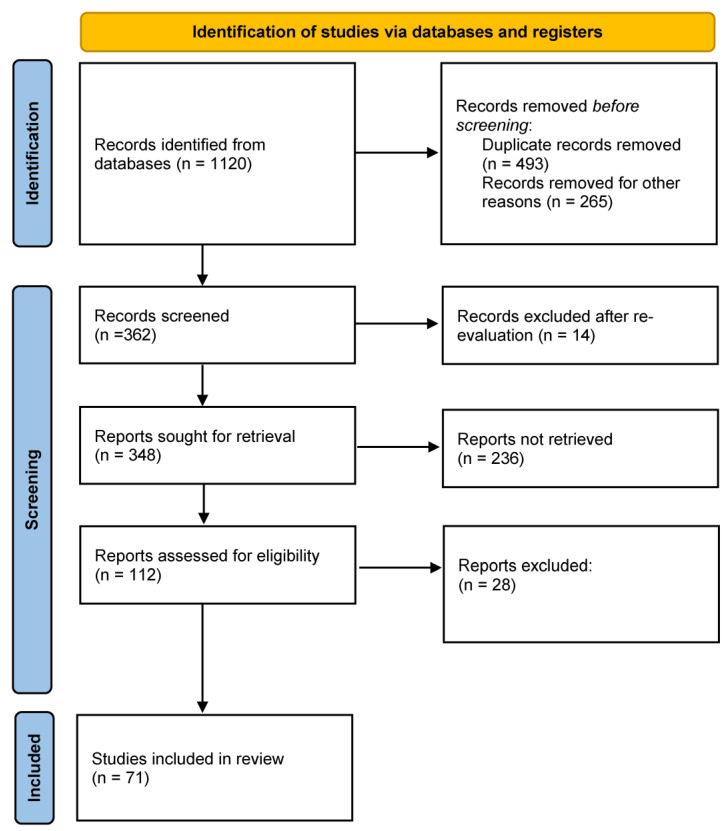
Process of identification and inclusion of studies.

**Table 1 jcm-12-00526-t001:** An overview of the treatment options.

Overview of the Treatment Options in Pelvic Pain						
Pharmacological treatment	Nonopioid analgesics	Tramadol	Opioid analgesics	Antidepressants	Anticonvulsants	Skeletal muscle relaxants
	NSAIDs have 3 desirable pharmacological effects: anti-inflammatory, analgesic and antipyretic. All NSAIDs and COX-2 appear to be equally effective in treating pain disorders.	Tramadol is a dual-action drug: one-third of its action is due to an opioid-like mechanism and two-thirds is due to an amitriptyline-like mechanism. It is a multimodal drug very useful in the treatment of pain.	The largest group of available opioids are µ-opioid receptor agonists or drugs with direct affinity for the µ-opioid receptor. Pure agonists have no apparent ceiling effect for analgesia.	They are a group of co-analgesics whose side effects appear early, and the analgesic effect may occur only after a few weeks. Patients should be informed about the rationale for the use of antidepressants and the fact that they are not treated as if they were suffering from psychological problems and about the prolonged waiting time for analgesic effect.	They are useful in chronic nociceptive pain, especially when the pain is described as stabbing or burning.	Most muscle relaxants are approved by the FDA for the treatment of spasticity (baclofen, dantrolene, and tizanidine) or musculoskeletal conditions (carisoprodol, chlorzoxazone, cyclobenzaprine, metaxalone, methocarbamol, and orphenadrine) while also exhibiting pain relief.
Microbiota	Orally administered selected bacterial or yeast cultures, most often lactic acid bacteria, whose task is to act in a beneficial way in the digestive tract, through immunomodulation and maintaining the normal physiological flora.					
Transcutaneous Electrical Nerve Stimulation (TENS)	Method of treating chronic pain. It can also be used in the acute stage of the disease, because it is not required to place the electrodes directly at the site of pain.					
Nerve Blocks	A method of treatment consisting in affecting the elements of the peripheral nervous system by injecting various solutions of anesthetics, primarily novocaine. A distinction is made between lumbar, cervical and ganglion blocks.					

## Data Availability

The datasets used and/or analyzed within the framework of this study are available from the corresponding author on reasonable request.

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
