# Peer review of "Management of Pelvic Pain in Patients with Crohn’s Disease—Current Overview"

_jcm, 2023, doi:10.3390/jcm12020526_

Round 1

Reviewer 1 Report

This is an exciting subject.  CD patients who suffer from fistule run into several problems during treatment.

1) Line 31 in the Introduction part need to have a reference . I prefer to add this reference."Amini Kadijani A, Javadinia F, Mirzaei A, Khazaei Koohpar Z, Balaii H, Baradaran Ghavami S, Gholamrezaei Z, Asadzadeh-Aghdaei H. Apoptosis markers of circulating leukocytes are associated with the clinical course of inflammatory bowel disease. Gastroenterol Hepatol Bed Bench. 2018 Winter;11(Suppl 1):S53-S58. PMID: 30774808; PMCID: PMC6347989."

2) Microbiota plays a critical role in IBD. It is better to have a separate part explaining the role of microbiota in fistula formation.

Author Response

Authors would like to thank the Reviewer for valuable comments and suggestions.

Suggested reference has been added. 

We provided a fragment about the role of microbiota in IBD and in perianal fistula in course of Crohn disease.

We hope our revised and corrected paper will meet your expectations.

Reviewer 2 Report

It is an interesting, documented, and complex paper about the etiopathogeny and management of pelvic pain in Crohn’s disease (CD) patients.

The article has a lot of information but sometimes it is difficult to follow the text. So, I suggest at least 1 figure (for example, a summary of the pathogenesis of pelvic pain in CD) and 1 table (e.g., an overview of the treatment options) make the text easier to understand.  

I suggest also discussing cannabis – known to have a therapeutic potential in IBD.  

In my opinion, the methodology should be presented at the beginning of the article (after the introduction and aim). The references related to treatment should be highlighted in the Table.

There are also some minor errors:

·       In the paragraph between lines 75 - 87 there is a lot of information but no bibliographic reference;

·       There are abbreviations without explanations (eg, TRPV, JAK);

·       The subtitles writing needs to be revised (e.g.,” Management of pelvic pain in Crohn’s disease” should be written with bold characters).

Author Response

Authors would like to thank the Reviewer for valuable comments and suggestions.

The article has a lot of information but sometimes it is difficult to follow the text. So, I suggest at least 1 figure (for example, a summary of the pathogenesis of pelvic pain in CD) and 1 table (e.g., an overview of the treatment options) make the text easier to understand.  
Answer: We have provided suggest figure and table (table is upload as separate Word file).

I suggest also discussing cannabis – known to have a therapeutic potential in IBD.  
Answer: We have added a paragraph about therapeutic potential of Cannabis in IBD.

In my opinion, the methodology should be presented at the beginning of the article (after the introduction and aim). The references related to treatment should be highlighted in the Table.
Answer: We have prepared manuscript according to the temple provide by the MDPI. If you find reorganizing the paper's section we will do it.

       In the paragraph between lines 75 - 87 there is a lot of information but no bibliographic reference;
Answer: References have been provided.

  There are abbreviations without explanations (eg, TRPV, JAK);
Answer: Abbreviations have been explained.

The subtitles writing needs to be revised (e.g.,” Management of pelvic pain in Crohn’s disease” should be written with bold characters).
Answer: The subtitles have been revised accordingly to your suggestion.

We hope our revised and corrected paper will meet your expectations.